# ROBUST REINFORCEMENT LEARNING FOR AUTONOMOUS DRIVING

**Yesmina Jaafra**[1,2,3]**, Jean Luc Laurent** [1]**, Aline Deruyver** [2] **& Mohamed S. Naceur** [3]
`yasmina.jaafra@etu.unistra.fr, jeanluc.laurent@segula.fr`
`aline.deruyver@unistra.fr, naceurs@yahoo.fr`
[1]: Segula Technologies, Parc d'activité de Pissaloup - Trappes, France
[2]: ICube Laboratory, Strasbourg University, France
[3]: LTSIRS Laboratory, ENIT, Tunisia

## ABSTRACT

Autonomous driving is still considered as an "unsolved problem" given its inherent important variability and that many processes associated with its development like vehicle control and scenes recognition remain open issues. Despite reinforcement learning algorithms have achieved notable results in games and some robotic manipulations, this technique has not been widely scaled up to the more challenging real world applications like autonomous driving. In this work, we propose a deep reinforcement learning ($RL$) algorithm embedding an actor critic architecture with multi-step returns to achieve a better robustness of the agent learning strategies when acting in complex and unstable environments. The experiment is conducted with Carla simulator offering a customizable and realistic urban driving conditions. The developed deep actor $RL$ guided by a policy-evaluator critic distinctly surpasses the performance of a standard deep $RL$ agent.

## 1 INTRODUCTION

An important approach for goal-oriented optimization is reinforcement learning ($RL$) inspired from behaviorist psychology (Sutton & Barto, 2018). The frame of $RL$ is an agent learning through interaction with its environment driven by an impact (reward) signal. The environment return reinforces the agent to select new actions improving learning process, hence the name of reinforcement learning (Jaafra et al., 2018). $RL$ algorithms have achieved notable results in many domains as games (Mnih et al., 2015; Silver et al., 2016) and advanced robotic manipulations (Levine et al., 2016; Lillicrap et al., 2016) beating human performance. However, standard $RL$ strategies that randomly explore and learn faced problems lose efficiency and become computationally intractable when dealing with high-dimensional and complex environments(Wahlström et al., 2015).

Autonomous driving is one of the current highly challenging tasks that is still an "unsolved problem" more than one decade after the promising 2007 DARPA Urban Challenge (Buehler et al., 2009). The origin of its difficulty lies in the important variability inherent to the driving task (e.g. uncertainty of human behavior, diversity of driving styles, complexity of scene perception...).

In this work, we propose to implement an advantage actor-critic approach with multi-step returns for autonomous driving. This type of $RL$ has demonstrated good convergence performance and faster learning in several applications which make it among the preferred $RL$ algorithms (Grondman et al., 2012). Actor-critic $RL$ consolidates the robustness of the agent learning strategy by using a temporal difference ($TD$) update to control returns and guide exploration. The training and evaluation of the approach are conducted with the recent CARLA simulator (Dosovitskiy et al., 2017). Designed as a server-client system, where the server runs the simulation commands and renders the scene readings in return, CARLA is an interesting tool since physical autonomous urban driving generates major infrastructure costs and logistical difficulties. It particularly offers a realistic driving environment with challenging properties variability as weather conditions, illumination, and density of cars and pedestrians.

The next sections review previous work on actor-critic $RL$ and provide a detailed description of the proposed method. After presenting CARLA simulator and related application advantages, we evaluate our model using this environment and discuss experimental results.

## 2 RELATED WORK

Various types of $RL$ algorithms have been introduced and are classified into three categories, actor, critic or actor-critic depending on whether they rely on a parameterized policy, a value function or a combination of both to predict actions (Konda & Tsitsiklis, 2003). In the actor-only methods, a gradient is generated to update the policy parameters in a direction of improvement (Williams, 1992). Despite policy gradients offer tough convergence guarantees, they may suffer from high variance resulting in slow learning (Berenji & Vengerov, 2003). On the other hand, critic-only methods built on value function approximation, use $TD$ learning and show lower variance of estimated returns (Boyan, 2002). However, they lack reliable guarantee of converging and reaching the real optimum (Grondman et al., 2012).

Actor-critic methods combine the advantages of the two previous ones by inducting a repetitive cycle of policy evaluation and improvement. Barto et al. (1990) is considered as the starting point that defined the basics of actor-critic algorithms commonly used in recent research. Since then, several algorithms have been developed with different directions of improvements. Wang et al. (2007), introduced the Fuzzy Actor-Critic Reinforcement Learning Network (FACRLN), which involves one neural network to approximate both the actor and the critic. Based on the same strategy, Niedzwiedz et al. (2008) developed the Consolidated Actor-Critic Model (CACM). Jan et al. (2003) used for the first time a natural gradient (Amari & Douglas, 1998) for the policy updates in their actor-critic algorithm. Silver et al. (2014) presented the Deterministic Policy Gradient algorithm (DPG) that assign a learned value estimate to train a deterministic policy. Recently, Mnih et al. (2016) proposed the Asynchronous Advantage Actor-Critic (A3C) algorithm where multiple agents operate in parallel allowing data decorrelation and learning experience diversity.

Despite that several actor-critic methods have been developed, most of them were tested on standard $RL$ benchmarks. The latter generally include basic tasks with low-level complexity comparatively to real world applications, like cart-pole balancing (Wang et al., 2007; Jan et al., 2003), maze problems (Niedzwiedz et al., 2008), multi-armed bandit (Silver et al., 2014), Atari games (Mnih et al., 2016; Gruslys et al., 2018) and OpenAI Gym tasks (Parisi et al., 2019; Lillicrap et al., 2016). Our work contribution consists in extending actor-critic $RL$ application to a very challenging task which is urban autonomous driving. The domain setting is particularly difficult to handle due to intricate and conflicting dynamics. Indeed, the driving agent must interact, in changing weather and lighting conditions and through a wide action space, with several actors that may behave unexpectedly, identify traffic rules and street lights, estimate appropriate speed and distance...

Our approach, that will be detailed in the next section, incorporates an actor and a multi-step $TD$ critic component to improve the stability of the $RL$ method.

## 3 ADVANTAGE ACTOR CRITIC WITH MULTI-STEP RETURNS

The $RL$ task considered in this work is a Markov Decision Process (MDP) $T_i$ defined according to the tuple $(S, A, p, r, \gamma, \rho_0, H)$ where $S$ is the set of states, $A$ is the set of actions, $p(s_{t+1}|s_t, a_t)$ is the state transition distribution predicting the probability to reach a state $s_{t+1}$ in the next time step given current state and action, $r$ is a reward function, $\gamma$ is the discount factor, $\rho_0$ is the initial state distribution and $H$ the horizon. Consider the sum of expected rewards (return) from a trajectory $\tau_{(0,H-1)} = (s_0, a_0, ..., s_{H-1}, a_{H-1}, s_H)$. A $RL$ setting aims at learning a policy $\pi$ of parameters $\theta$ (either deterministic or stochastic) that maps each state $s$ to an optimal action $a$ maximizing the return $R$ of the trajectory.

$$R_t = r_{t+1} + \gamma R_{t+1} = \sum_{i=t}^{t+H-1} \gamma^{i-t} r_{i+1} \tag{1}$$

Following the discounted return expressed above, we can define a state value function $V(s) : S \to R$ and a state-action value function $Q(s, a) : A \times S \to R$ to measure, respectively, the current state and state-action returns estimated under policy $\pi$:

$$V(s_t) = \mathbb{E}[R_t | s_t = s] \tag{2}$$

$$Q(s_t, a_t) = \mathbb{E}[R_t | s_t = s, a_t = a] \tag{3}$$

In value-based $RL$ algorithms such as Q-learning, a value function is approximated to select the best action according to the maximum value attributed to each state and action pair. On the other hand, policy-based methods directly optimize a parameterized policy without using a value function. They use instead gradient descents like in the family of REINFORCE algorithms (Williams, 1992) updating the policy parameters $\theta$ in the direction:

$$\Delta\theta = \alpha\nabla_\theta \log \pi_\theta(s_t | a_t) R_t \tag{4}$$

The main problem with policy based methods is that the score function $R_t$ uses the averaged rewards calculated at the end of a trajectory which may lead to the inclusion of "bad" actions and hence slow learning. The solution provided in actor-critic framework is to replace the reward function $R_t$ in the policy gradient (equation 4) with the action value function that will enable the agent to learn the long-term value of a state and therefore enhance its prediction decision:

$$\Delta\theta = \alpha\nabla_\theta \log \pi_\theta(s_t | a_t) Q(s_t, a_t) \tag{5}$$

Then train a critic to approximate this value function parameterized with $\omega$ and update the model accordingly. At this point, we can conclude that an efficient way to derive an optimal control of policies is to evaluate them using approximated value functions. Hence, building accurate value function estimators results in better policy evaluation and faster learning.

$TD$ learning combining Monte Carlo method and dynamic programming (Sutton & Barto, 2018) has proved to be an effective way to calculate good approximations of value functions by allowing an efficient reuse of rewards during policy evaluation. It consists in taking an action according to the policy and bootstrapping the 1-step sampled return from the value function estimate resulting in the below 1-step $TD$ target:

$$G_t = r_t + \gamma * V_t(s_{t+1}) \tag{6}$$

Given the last return estimation, we obtain the 1-step $TD$ update rule that allows the adjustment of the value function according to the $TD$ error $\delta_t$ with step size $\beta$:

$$V(s_t) = V(s_t) + \beta(\underbrace{r_t + \gamma V_t(s_{t+1}) - V(s_t)}_{\delta_t}) \tag{7}$$

At this level, the actor-critic algorithm still suffers from high variance. In order to reduce the variance of the policy gradient and stabilize learning, we can subtract a baseline function, e.g. the state value function, from the policy gradient. For that, we define the advantage function $A(s_t, a_t)$ which calculates the improvement in predicting an action compared to the average $V(s_t)$:

$$A(s_t, a_t) = Q(s_t, a_t) - V(s_t) \tag{8}$$

An approximation of the advantage function is required since it involves two value functions $Q(s_t, a_t)$ and $V(s_t)$. Therefore let's reformulate $A(s_t, a_t)$ as the difference between the expected future reward and the actual reward that the agent receives from the environment (Heess et al., 2013):

$$A(s_t, a_t) = R(s_t, a_t) - V(s_t) \tag{9}$$

When used in the previous policy gradient (equation 5), this gives us the advantage of the actor policy gradient:

$$\Delta\theta = \alpha\nabla_\theta \log \pi_\theta(s_t|a_t)(G_t - V(s_t)) \tag{10}$$

We can subsequently assume that $TD$ error is a good candidate to estimate the advantage function. Accordingly, we deduce the final actor policy gradient:

$$\Delta\theta = \alpha\nabla_\theta \log \pi_\theta(s_t|a_t)\delta_t \tag{11}$$

Given the complex nature of the autonomous urban driving task, we will use a generalized version of $TD$ learning by extending the bootstrapping over multiple time steps into the future. Algorithmically, we will define configurable multi-step returns within the $TD$ target. Hence, $TD$ error becomes:

$$\delta_t = [\sum_{i=t}^{t+H-1} \gamma^{i-t}r_i] + \gamma^H V(s_{t+H}) - V(s_t) \tag{12}$$

Multi-step returns have been demonstrated to improve the performance of learning especially with the advent of deep $RL$ (Mnih et al., 2016). Indeed, it allows the agent to gather more information on the environment before calculating the error in the critic estimates and updating the policy.

So far, we have a good theoretical basis to launch our agent. The experiments carried out by the application of this approach in the Carla simulator will be presented in the next section.

## 4 EXPERIMENT

In this section we investigate the performance of an advantage actor-critic (A2C) algorithm embedding multi-step $TD$ target updates on the challenging task of urban autonomous driving. The goal of our experimental evaluation is to demonstrate that the incorporation of a multi-step returns critic (MSRC) component in a deep $RL$ framework consolidates the robustness of the agent by controlling and guiding its learning strategy. We expect a reduction of the actor gradient variance, an ascendant trend of episodic average returns and more generally a better performance comparatively to the case where the MSRC component is deactivated in the A2C algorithm.

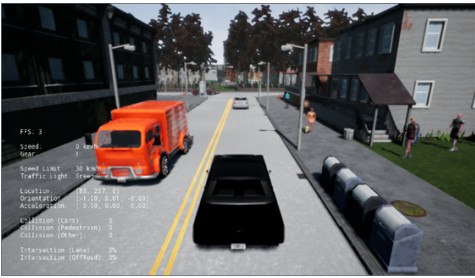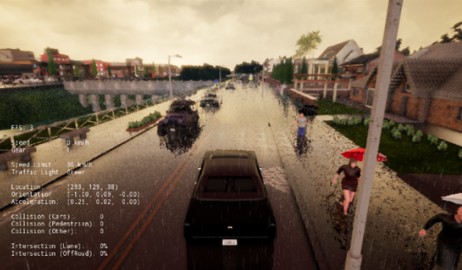

Figure 1: Carla environments. Left: Clear Noon weather in Town 2. Right: Hard Rainy in Town 1.

**Environment.** We conduct the experiments using CARLA simulator for autonomous driving which provides an interesting interface allowing our $RL$ agent to control a vehicle and interact with a dynamic environment. Comparatively to existing platforms, Carla offers a customizable and quite realistic urban driving conditions with a set of advanced features for controlling the vehicle and gathering the environment feedback. It is designed as a server-client system where the server implemented in Unreal Engine 4 (UE4) [1] runs the simulation commands and returns the scene readings.

---

[1] https://www.unrealengine.com

The client implemented in Python sends the agent predicted actions mapped as driving commands and receives the resulting simulation measures that will be interpreted as the agent rewards.

Carla $3D$ environment consists of static objects as buildings, roads and vegetation and dynamic non-player characters, mainly pedestrians and vehicles. During training, we can episodically vary server settings as the traffic density (number of dynamic objects) and visual effects (weather and lightening conditions, sun position, cloudiness, precipitation...). Some examples of resulting environments are illustrated in figure 1.

**Observation and action spaces.** The agent interacts with the environment by generating actions and receiving observations over regular time steps. The action space selected for our experiments is built on the basis of three discrete driving instructions (steering, throttle, and brake) extended with some combinations in-between (turn left and accelerate/decelerate...). The observation space includes sensors outputs as color images produced by RGB cameras and derived depth and semantic segmentations. The second type of available observations consists in a range of measurements reporting the vehicle location (similarly to GPS) and speed, number of collisions, traffic rules and positioning of non-player dynamics characters.

**Rewards.** A crucial role is played by rewards in building driving policies as they orient the agent predictions. In order to further optimal learning, the reward is shaped as a weighted sum of measurements extracted from the observations space described in the previous paragraph. The idea is to compute a difference between the current ($step\ t$) and the previous ($step\ t-1$) measure of the selected observation then impact it positively or negatively on the aggregated reward. The positively weighted variables are distance traveled to target and speed in km/h. The negatively weighted variables are collisions damage (including collisions with vehicles, pedestrians and other), intersections with sidewalk and opposite lane. For example, the agent will get a reward if the distance to goal decreases and a penalty each time a collision or an intersection with the opposite lane is recorded.

**Experiment settings.** The agent training follows a goal-directed navigation on straight roads from scratch. An episode is terminated when the target destination is reached or after a collision with a dynamic non-player character. The A2C networks are trained with 10 millions steps for 72 hours of simulated continuous driving. Motivated by the recent success achieved by deep $RL$ in challenging domains (Mnih et al., 2016), we use convolutional neural networks (CNN) to approximate both the value function of the critic and the actor policy where the parameters are represented by the deep network weights.

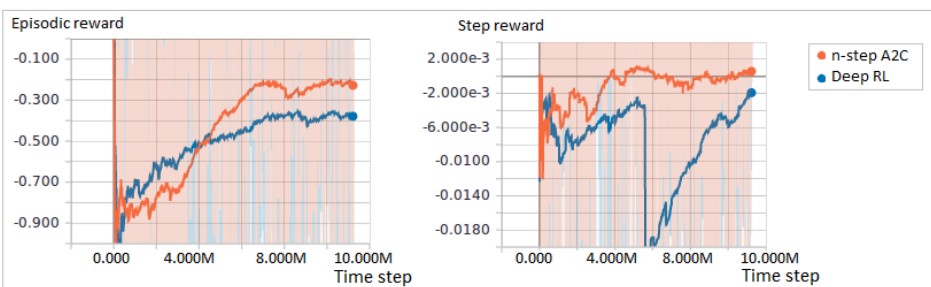

Figure 2: Training phase - Comparison between n-step A2C and standard deep $RL$ performance trained in Town 2.

The CNN architectures consist of $4$ convolutional layers, $3$ max-pooling layers and one fully connected layer at the output. The discount factor is set as $0.9$. We used $10$-step rollouts, with initial learning rate set as $0,0001$. Learning rate is linearly decreased to zero over the course of training. While training the approach, a stochastic gradient descent is operated each $10$ time steps and the resulting policy model is stored only if its performance (accumulated rewards) exceeds the last retained model. The final stored model is then used in the test phase.

**Comparative evaluation.** In the absence of various state-of-the-art works on the recent CARLA simulator, we choose to compare 2 versions of our algorithm: the original deep actor $RL$ guided by the MSRC policy-evaluator versus a standard deep actor $RL$ resulting from the deactivation of the MSRC component in the original algorithm. In fact the few available state-of-the-art results in

CARLA environment (Dosovitskiy et al., 2017; Liang et al., 2018) report the percentage of successfully completed episodes. This type of quantitative evaluation doesn't meet our experiment objectives mentioned in the beginning of this section to evaluate and interpret the MSRC contribution in complex tasks like autonomous driving. Guided by the several works on $RL$ strategies in different domains (Mnih et al., 2016), (Parisi et al., 2019), we selected episodic average and cumulative rewards metrics to evaluate our approach.

Figure 2 shows the generated reward in training phase. We use average episodic reward to describe the methods global performance and step reward to emphasize the predictions return variance. We can make few observations in this regard. In term of performance, our n-step A2C approach is dominant over almost all the 10000 training episodes confirming the efficiency of the $RL$ strategy controlled by the MSRC. Furthermore, we noticed that regarding the best retained models, the A2C stored just few models (5) in the 2000 first episodes, then this number drastically increased to 100 retained models in the remaining 8000 episodes. This means that our method early achieved the exploration phase and moved to exploitation from the training level of 2000 episodes. On the other hand, the standard deep $RL$ totalized only 10 best models over the training phase reflecting the weak efficiency of a random strategy to solve a very complex and challenging problem like autonomous driving. A last visual interpretation that we can deduce from the step reward graph is that the variance of A2C predictions is significantly reduced relatively to the standard deep $RL$ confirming the $TD$ learning contribution in accomplishing a faster learning.

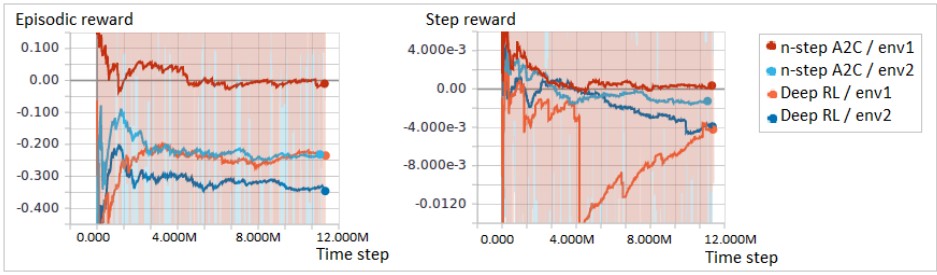

Figure 3: Testing Phase - Evaluation of n-step A2C and standard deep $RL$ tested in 2 different environments env1 and env2. (Both have been trained in env1).

Figure 3 recaps the testing phase evaluation following two different scenarios. First, the testing was conducted in the same environment and conditions as the training: Town 2 and Clear Noon weather (env1). From the episodic reward graph we can observe that our approach substantially outperforms the standard deep $RL$ which means that training with multi-step returns critic leads to more efficient $RL$ models. In the second scenario, both methods agents are tested in a different environment than training: Town 1 and in hard rainy conditions (env2). The n-step A2C is still more competitive than the standard deep $RL$ showing superior generalization capabilities in the new unseen setting. Nevertheless, its performance has decreased in the second test scenario reflecting a certain fragility to changing environment. On the other side, the standard deep $RL$ is still showing higher prediction return variance in the step reward graph confirming training phase conclusions.

## 5  CONCLUSION

In this paper we addressed the limits of $RL$ algorithms in solving high-dimensional and complex tasks. Combining both actor and critic methods advantages, the proposed approach implemented a continuous process of policy assessment and improvement using multi-step $TD$ learning. Evaluated on the challenging problem of autonomous driving using CARLA simulator, our deep actor-critic algorithm demonstrated higher performance and faster learning capabilities than a standard deep $RL$. Furthermore, the results showed a certain vulnerability of the approach when facing unseen testing conditions. Considering this paper as a preliminary attempt to scale up $RL$ approaches to high-dimensional real world applications like autonomous driving, we plan in future work to examine the performance of other $RL$ methods such as deep Q-learning and Trust Region Policy Optimization (Schulman et al., 2015) on similar complex tasks. Furthermore, we propose to tackle the issue of non-stationary environments impact on $RL$ methods robustness as a multi-task learning problem

(Caruana, 1998). In such context, we will explore recently applied concepts and methodologies such as novel adaptive dynamic programming (ADP) approaches, context-aware and meta-learning strategies. The latter are currently attracting a keen research interest and particularly achieving promising advances in designing generalizable and fast adapting $RL$ algorithms (Santoro et al., 2016; Ravi & Larochelle, 2017). Subsequently, we will be able to increase driving tasks complexity and operate conclusive comparisons with the few available state-of-the-art experiments on CARLA simulator.

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
