# OpenReview forum: "Robust Reinforcement Learning for Autonomous Driving "
_ICLR.cc/2019/Workshop/drlStructPred — drlStructPred 2019_

### Official Review · AnonReviewer1 · 2019-03-31
**Experimental paper with unclear contribution**

**Rating:** 2
**Confidence:** 2

**Review:**

This paper tackles the problem of autonomous driving using deep RL algorithms. More specifically, the authors evaluate the benefit of using a multi-step returns critic in A2C for this task. Experiments are conducted using the realistic driving simulator CARLA.

Very limited contribution:
* Though investigating deep RL approaches on the difficult task of autonomous driving, I find the experiments to be very limited as only one algorithm (A2C) is considered. It would have been interesting to present a broader study with more than one method.
* Dosovitskiy et al. (2017) already evaluate A3C for the autonomous driving task using CARLA. The contribution of the current paper therefore seems limited to evaluating the benefit of multi-step returns.

It is not clear how rewards are defined:
* What are the weights given to each "goal feature"?
* How were these weights chosen?
* Would a "good policy" in terms of those rewards actually be considered "good" by humans?

Experiments:
* How many repetitions were performed?
* Does the shaded are on Figs. 2-3 correspond to standard deviation? If so, could you really conclude that there was a difference between the two compared methods?
* All experiments are performed on straight roads. It would be interesting to see how different/similar results are on more challenging roads.
* What were the environmental conditions during training? If it was always sunny, one cannot really be surprised that the methods do not generalize to different weathers...
* Comparison with state-of-the-art results? I understand that existing approaches evaluate performance differently in their papers. The approaches could still be run on the given setting.

The paper is easy to read. The contribution is presented as a study of deep RL techniques for autonomous driving, which is relevant for the workshop. However, this has already been done in the past (e.g. Dosovitskiy et al., 2017), especially using algorithms very close to what is considered here, and the approaches studied previously were not included in the current paper. Moreover, the experiments lack details to actually make the presented comparison meaningful.

Minor comments:
* [Eq.4] R(t) should be R_t.
* When citing multiple references one after the other, putting them in the same parenthesis increases readability.

---

### Official Review · AnonReviewer3 · 2019-04-04
**Nice task but it should have more experiments.**

**Rating:** 3
**Confidence:** 2

**Review:**

This paper evaluates two common RL algorithms in a simulated driving environment.

Pros:
1- Nice choice of using a more realistic environment.
2- The paper is easy to read and understand.

Cons:
1- Some details are missing: how exactly the reward is computed? How long an episode lasts, on average? What is the average distance a vehicle runs before having a collision?
2- Only two RL methods are evaluated (A2C and A3C). Given the abundance of methods in the literature, it would be nice to see comparisons with some other commonly used methods, such as Q-Learning and TRPO.

Questions:
1- Does the agent use a third person (outside the vehicle) camera view? If yes, the results would be more convincing if the first-person view was used because that is closer to the viewpoint of real vehicle.
2- In section 4 you mention that traffic rules are given as input to the agent. How do you extract these rules from the environment? For example, how a stop sign is given as an input to the agent? Also, a more realistic environment should not explicitly give these to the agent. Instead, inferring traffic rules should be a job of the agent as traffic signs often change due to constructions, accidents, etc.

---

### Official Review · AnonReviewer5 · 2019-04-05
**Low improvement, with little evidence.**

**Rating:** 1
**Confidence:** 2

**Review:**

The paper struggles to clearly explain their contribution to current works, and improvements over a baseline. The written quality of the paper shows numerous grammatical errors and careless mistakes. On a positive note they do demonstrate the variance improvement of using multistep rewards over a single step look ahead.

---

### Official Review · AnonReviewer4 · 2019-04-07
**Innovation and contribution**

**Rating:** 1
**Confidence:** 2

**Review:**

 This work is about autonoous driving and proposes a deep reinforcement learning (RL) algorithm embedding an actor critic architecture with multi-step returns to achieve a better robustness of the agent learning strategies when acting in complex and unstable environments.

The major problem of this work is that its technical/research innovation is not clear. Actually the proposed algorithm is not new and similar or more advanced algorithms have already been proposed, such as Generalized Advantage Estimation, Schulman et al. 2016(b). This paper is more like a homework in a RL course.

---

### Decision · Program_Chairs · 2019-04-08
**Acceptance Decision**

**Decision:**

Accept

**Comment:**

Even though the results are very preliminary we still accept them for the purpose of fostering interesting discussions.